

# The role of Rnf in ion gradient formation in *Desulfovibrio alaskensis*

Luyao Wang[1], Peter Bradstock[1], Chuang Li[1], Michael J. McInerney[1] and Lee R. Krumholz[1,2]

[1] Department of Microbiology and Plant Biology, University of Oklahoma, Norman, Oklahoma, USA
[2] Institute for Energy and the Environment, University of Oklahoma, Norman, Oklahoma, USA

## ABSTRACT

Rnf is a membrane protein complex that has been shown to be important in energy conservation. Here, *Desulfovibrio alaskensis* G20 and Rnf mutants of G20 were grown with different electron donor and acceptor combinations to determine the importance of Rnf in energy conservation and the type of ion gradient generated. The addition of the protonophore TCS strongly inhibited lactate-sulfate dependent growth whereas the sodium ionophore ETH2120 had no effect, indicating a role for the proton gradient during growth. Mutants in *rnfA* and *rnfD* were more sensitive to the protonophore at 5 µM than the parental strain, suggesting the importance of Rnf in the generation of a proton gradient. The electrical potential ($\Delta\Psi$), $\Delta$pH and proton motive force were lower in the *rnfA* mutant than in the parental strain of *D.alaskensis* G20. These results provide evidence that the Rnf complex in *D. alaskensis* functions as a primary proton pump whose activity is important for growth.

The Rnf complex was first discovered in *Rhodobacter capsulatus* and is thought to be transcribed in one cluster of seven genes *rnf*ABCDGEH (*Jouanneau et al., 1998*). Evidence that depleting one subunit could destabilize others indicated the formation of a complex (*Kumagai et al., 1997*) which was later shown to be a protein complex involved in the *Rhodobacter* nitrogen fixation process (*Schmehl et al., 1993*). The Rnf complex has been differentiated into three major groups based on gene organization (*Biegel et al., 2011*). The first group found in *R. capsulatus*, *Pseudomonas stutzeri* (*Yan et al., 2008*) and *Azotobacter vinelandii* (*Curatti et al., 2005*) with *rnf*ABCDGE. Another group with the gene order *rnf*CDGEAB is found in *A.woodii* (*Biegel, Schmidt & Müller, 2009*) and *Clostridium kluyveri* (*Seedorf et al., 2008*). Lastly *rnf*BCDGEA is found in *Chlorobium limicola*, *Bacteroides vulgatus* and *Prosthecochloris aestuarii* (*Biegel et al., 2011*).

Rnf complex has been shown to be a novel type of ferredoxin-dependent enzyme, catalyzing the oxidation–reduction reaction between reduced ferredoxin and $NAD^+$ (*Biegel, Schmidt & Müller, 2009*; *Boiangiu et al., 2005*). Because of the high sequence similarity to the $Na^+$-translocating NADH:ubiquinone oxidoreductase (Nqr) (*Kumagai et al., 1997*), the Rnf complex in *R. capsulatus* was suggested to be involved in ion translocation using the energy from the exergonic reduction of $NAD^+$ coupled to the oxidation of reduced ferredoxin (*Müller et al., 2008*). Based on the above similarity, Rnf was originally proposed

Corresponding author
Lee R. Krumholz, krumholz@ou.edu

to be a sodium pump (*Biegel, Schmidt & Müller, 2009*). This theory was strengthened after the Rnf complex in *Acetobacterium woodii* was confirmed to generate a $Na^+$ gradient when carrying out the redox reaction between reduced ferredoxin and $NAD^+$ (*Biegel & Müller, 2010*). However, the Rnf complex was suggested to be a proton-translocating ferredoxin:$NAD^+$ oxidoreductase in *Clostridium ljungdahlii* (*Kopke et al., 2010*) and was more recently shown to produce a proton gradient (*Tremblay et al., 2013*). It therefore appears that Rnf may pump different ions in different bacteria (*Hess et al., 2016*).

*Desulfovibrio alaskensis* G20 encodes the Rnf complex with a unique gene arrangement (*rnf*CDGEABF); however, a similar Rnf complex is present in most sulfate-reducing bacteria for which genome sequences are available (*Pereira et al., 2011*). Directly preceding the operon is a decaheme cyctochrome (dhcA) which is also present in many but not all sulfate reducing bacteria. This decaheme cytochrome belongs to cytochrome $c_3$ family (*Pereira et al., 2011*). The final gene in the operon, *rnf* F is most similar to *apb*E, a membrane associated lipoprotein (*Beck & Downs, 1999*).

Recently, it was reported that mutants in Rnf in *D. alaskensis* are unable to grow using $H_2$ or during syntrophic growth conditions with *Syntrophus aciditrophicus* or *Syntrophomonas wolfeii* (*Krumholz et al., 2015*; *Price et al., 2014*). By investigating gene expression levels of *D.alaskensis* G20 grown in pure culture or under syntrophic conditions (*Krumholz et al., 2015*), it was suggested that both formate and $H_2$ can act as electron shuttles during syntrophic growth. Those experiments also showed that expression of *rnf* genes are upregulated under syntrophic conditions and during $H_2$ dependent growth confirming the role of the Rnf complex in $H_2$ metabolism.

In this experiment, mutants of RnfA (Dde_0585) and RnfD (Dde_0582) were studied and compared with the G20 parent strain to determine whether Rnf in *D. alaskensis* is involved in generation of a proton motive force (PMF). These two mutants of *Desulfovibro alaskensis* G20 were obtained by using mini Tn10 transposon (*Groh et al., 2005*) and identified individually during syntrophic growth with *Syntrophomonas wolfeii* (*Krumholz et al., 2015*).

## MATERIALS AND METHODS
### Growth of cultures
*Desulfovibrio alaskensis* G20 and *rnf* mutants were routinely cultured anaerobically ($N_2$-$CO_2$, 80:20) in basal medium with lactate-sulfate (50 mM each) as the electron donor and electron acceptor, respectively and containing 0.1% yeast extract (*Krumholz et al., 2015*). The medium was prepared under a headspace of $N_2$/$CO_2$(80/20, vol/vol) with sodium bicarbonate (3.5 g/L) added as a buffer. Before inoculation, the medium was reduced with 0.025% each cysteine and sulfide. Kanamycin (1050 μg/mL) was added to the mutant cultures. Inoculum consisted of 0.2 ml transferred into 10 ml of media or the same ratio for large volume cultures. Cultures were grown in the incubator at 37 °C and growth of triplicate cultures was measured by optical density at 600 nm. The stock cultures were frozen in 20% glycerol at −20°. Growth of syntrophic cocultures was as previously described (*Krumholz et al., 2015*).

**Table 1  List of primers used for gap analysis in the *rnf* operon.**

| Gap | Forward primer | Reverse primer |
|---|---|---|
| Gap CC (*cytC/rnfC*) | GGCATTCACGGCCTGTGCCT | CTTGTCGCCGGGGTGATCGG |
| Gap CD (*rnfC/rnfD*) | CCTGCTGGGACGCTACAGCG | ATGCCGTGTATGGTGCGCCC |
| Gap DG (*rnfD/rnfG*) | GGCAAGGCCGCCATGGTCAT | TGTACTCGGCCCCCGTGGAG |
| Gap GE (*rnfG/rnfE*) | ATCGGCTGCATGGTTGCCGT | ATTGGCGGACTTGGTGACCGC |
| Gap EA (*rnfE/rnfA*) | GGGCTTGTTCTGGGCGCCAT | CCGCCCATGCCCAGACTGAC |
| Gap BE (*rnfB/rnfF*) | AAAGCCTGCCTCGCGTTCGG | AAGCCCCAGCAGACCGCATG |

Growth was tested with a variety of other electron donors and acceptors: lactate (50 mM), pyruvate (25 mM), ethanol (10 mM) and formate (50 mM). Sulfate was used at 50 mM with lactate and 25 mM with other electron donors and sulfite was added at 10 mM.

The protonophore 3,3,4,5-tetrachlorosalicylanilide (TCS, 5 $\mu$M or 20 $\mu$M), or the sodium-specific ionophore N,N,N′,N′-Tetracyclohexyl-1,2-phenylenedioxydiacetamide (ETH2120, 20 $\mu$M) were added to media to test their effects on growth.

## Transcriptional analysis of *rnf* genes

Total RNA was extracted from the parent strain, *rnfA*, and *rnfD* mutants using the Qiagen RNeasy kit as previously described (*Krumholz et al., 2015*). Purity and adequate yield was confirmed with a diode-array spectrophotometer. First-strand cDNA synthesis was performed using the Fermentas Revertaid kit with gene-specific primers covering the entirety of the *rnf* mutants' operons. Primers were designed to span the gaps between each gene in the operon (Table 1). PCR was performed with the parent strain and mutants with each primer set followed by agarose gel analysis to determine whether all genes in the operon were expressed and whether the genes were on the same transcript.

### Quantitative PCR analysis

Triplicate cultures of the parent strain and *rnfA* and *rnfD* mutants were grown to mid-log phase (OD$_{600}$ 0.3–0.45) and harvested by centrifugation at 5,000 × g for 5 min at 4 °C. The pellet was homogenized in RNAprotect Bacteria reagent (Qiagen) for 5 min at room temperature to prevent degradation of RNA transcripts. Excess liquid was removed by centrifugation and cell pellets were resuspended in 200 ul of TE buffer containing 1 mg/ml lysozyme. Total RNA was extracted using the RNeasy mini kit (Qiagen). RNA was then treated with the Ambion Turbo DNA –free kit (Thermo) to eliminate genomic DNA contamination. RNA was quantified spectrophotometrically. cDNAs were synthesized using the First Stand cDNA synthesis kit (Fermentas) as described in the manual. Control PCR reactions were done to ensure there was no genomic DNA or non-specific amplification.

Quantitative PCR, was done in triplicate from each culture replicate using 10 ng cDNA as the template and the Maxima SYBR Green qPCR master mix (Thermo) with a MyIQ Cycler (Bio-Rad). Primers were designed using Primer-BLAST (NCBI website) to specifically amplify 128- to 155-bp regions of targeted genes Dde_587, Dde_589, Dde_ 590 (Table 2). Dde_ 0587 is the final gene in the *rnf* operon and the other two genes are the next two ORFs directly downstream of the operon. The 16S rRNA primers were as previously described (*Li et al., 2011*). Reactions used the following

**Table 2  Primers used for the RT-qPCR experiments.**

| Primer | Sequence | Target gene |
|---|---|---|
| Dde_ 587 forward | 5′-AACCGTGGGTTATCGCCATT-3′ | *rnfF* gene |
| Dde_ 587 reverse | 5′-ATGGCTGTACATGCGTTTGC-3′ | |
| Dde_ 589 forward | 5′-AAGGGCAGGTAGTGCTGATG-3′ | Putative regulator gene |
| Dde_ 589 reverse | 5′-GGTAAACTTCACGCCTTCGC-3′ | |
| Dde_ 590 forward | 5′-GCCTGCGGCTTAATTTCGAG-3′ | Cardiolipin synthase gene |
| Dde_ 590 reverse | 5′-ACGGTTTTCCAGTTCGCTCA-3′ | |
| 16S forward | 5′-ACGGTTGGAAACGACTGCTA-3′ | 16S rRNA gene |
| 16S reverse | 5′-AGCTAATCAGACGCGGACTC-3′ | |

amplification condition: 95 °C for 10 min; 40 cycles of denaturation at 95 °C for 15 s and annealing/extension at 60 °C for 1 min. mRNA expression was calculated using $(E_{target})^{\Delta Ct\_target(control-mutant)}/(E_{reference})^{\Delta Ct\_reference(control-mutant)}$ (*Pfaffl, 2001*). The 16S rRNA gene was used for normalization.

## Washed cell experiments

Washed cells were used to measure sulfide production under non-growth conditions. Briefly, 100 ml lactate-sulfate culture grown to mid log phase was harvested by centrifugation at 5,300 × g for 15 min, washed twice in buffer containing 50 mM MOPS (pH 7.2), 5 mM $MgCl_2$ and resuspended in the same buffer. All buffers were flushed with $N_2$ for 30 min. Assays were carried out in serum tubes containing 2 ml of buffer-cell mixture incubated at 37 °C on a shaker at 100 rpm. The assay mixture contained the washing buffer, 5 mM sodium sulfate and either 50 mM lactate, 50 mM sodium formate ($N_2$ headspace) or $H_2$ in the headspace. Between 50 and 200 µg of cell protein was added to each tube. Tubes were sacrificed by addition of 2 ml 10% Zinc Acetate at 60 min intervals for sulfide analysis. Sulfide was determined using the methylene blue assay (*Cline, 1969*).

## Measurement of proton motive force

Triplicate cultures of *D. alaskensis* parent strain and *rnf*A mutant were grown in lactate–sulfate basal medium until mid exponential phase ($OD_{600}$ of 0.5) in a 1 liter volume and 30 ml aliquots of cells were dispensed into 100 ml serum bottles under $N_2/CO_2$ (80/20, vol/vol). The basal medium contains 5 mM $K^+$, mainly added as potassium phosphate salts. Protein concentration was determined using the Bicinchoninic Acid Assay (Pierce BCA Protein Assay Kit).

Transmembrane electrical potential ($\Delta\Psi$) was measured as previously described (*Tremblay et al., 2013*). Briefly, cells were incubated with [$^3$H]tetraphenylphosphonium bromide ([$^3$H]TPP$^+$) (ARC, 0.1 µCi/ml) for 15 min at 37 °C (*Kashket & Barker, 1977*; *Shirvan, Schuldiner & Rottem, 1989*). [$^3$H]TPP$^+$ was dissolved in media and filtered through 0.45 µm filters prior to adding to cells. Controls were incubated with nigericin (ACROS ORGANICS, 20 µM dissolved in ethanol at 200×) and valinomycin (ACROS), 20 µM dissolved in DMSO at 200× for 15 min to eliminate the $\Delta\Psi$. The combination of nigericin (proton/$K^+$ antiporter) and valinomycin ($K^+$ uncoupler) will dissipate the proton gradient (*Kessler et al., 1977*). Cells were then separated from

the medium using silicone oil as described below and $[^3H]TPP^+$ was determined in the liquid scintillation counter (Packard TriCarb 2100TR). The uptake of $[^3H]TPP^+$ was corrected for extracellular contamination using the ratio of the intracellular to extracellular volume as described below. Non-specific binding of $[^3H]TPP^+$ was corrected by subtracting the cell associated $[^3H]TPP^+$ in the valinomycin/nigericin treatment from that in the untreated cells. The $\Delta\Psi$ was calculated with the simplified Nernst equation $(-2.3[RT/F] \times \log[(\text{concentration in})/(\text{concentration out})])$.

The $\Delta pH$ was measured by testing the distribution of $[^{14}C]$benzoate (ARC, $0.4\,\mu Ci/ml$) across the cell membrane after 15 min incubation at 37 °C with cells (*Kashket & Barker, 1977*; *Shirvan, Schuldiner & Rottem, 1989*). Cells were then separated from the supernatant as described below and $[^{14}C]$benzoate was measured. $[^{14}C]$benzoate uptake was corrected for extracellular contamination using the ratio of intracellular to extracellular volume described below. Tetrachlorosalicylanilide (TCS, $20\,\mu M$) was added to controls to eliminate the $\Delta pH$ and was used to calculate internal pH in the absence of a $\Delta pH$. The external pH was measured, $[^{14}C]$benzoate uptake was quantified and intracellular pH was calculated with the Henderson–Hasselbach equation (*Rottenberg, 1979*). The proton motive force (PMF) was calculated using the equation: $PMF = \Delta\Psi - z\Delta pH$. At 37 °C, z equals to 61.48.

## Measurement of intracellular volume/total volume of cell pellet

$^3H_2O$ ($1\,\mu Ci/ml$) was used to measure the total pellet volume while $[^{14}C]$taurine (PerkinElmer, $0.5\,\mu Ci/ml$) was used to measure extracellular volume and was inferred to penetrate up to the plasma membrane (*Kashket & Barker, 1977*). The intracellular water volume was estimated from total pellet water volume minus extracellular water volume by measuring the difference between the distribution of $^3H_2O$ and $[^3H]$taurine after incubating cells for 15 min at 37 °C with each isotope.

## Separation of cells from the supernatant

Following the 15 min incubation with the isotope, three 10 ml aliquots of the total 30 ml solution were put into three 15 ml Falcon tubes with 3 ml of a mixture of silicone oils (25% Fluid 510, 50 centistokes, and 75% Fluid 550, 115 centistokes; Dow-Corning Corp, Serva. vol/vol). Cells were then centrifuged at $5,300 \times g$ at 4 °C for 10 min. The aqueous layer and the silicone oil was removed with a Pasteur pipette connected to a vacuum line. The bottom of the falcon tube containing the cell pellet was cut off and moved into the scintillation vial and the cell pellet was re-suspended with 200 µl distilled-water. Liquid scintillation cocktail (Ultima Gold; PerkinElmer, Waltham, MA, USA) was added prior to scintillation counting.

## Statistical analysis

Statistical analysis used either the *t*-test or a one-way analysis of variance (ANOVA) with the Tukey's Range test.

# RESULTS

## Mutants in the *rnf* operon

In a previous study, a Tn10 mutant library of *D. alaskensis* G20 was screened for the ability to grow on butyrate in coculture with *Syntrophomonas wolfei* (*Krumholz et al., 2015*).

**Table 3** Normalized expression levels of genes within the *rnf* operon (from *Krumholz et al., 2015*). Expression was determined with the parent strain of *D. alaskensis* G20 during growth in pure culture with the indicated electron donors and acceptors and in coculture with *S. aciditrophicus* (SB) or with *S. wolfei* (SW) in the presence of benzoate or butyrate, respectively, as electron donors and sulfate as the electron acceptor.

| Gene | Function | Lactate/ Sulfate | $H_2$/ Sulfate | $H_2$/ Sulfite | SB | SW |
|------|----------|------------------|----------------|----------------|------|-------|
| Dde_ 0580 | cyt c family protein | 3.17 | 8.33 | 6.18 | 6.02 | 14.36 |
| Dde_ 0581 | RnfC subunit | 3.05 | 6.68 | 6.12 | 5.61 | 13.50 |
| Dde_ 0582 | RnfD subunit | 1.90 | 4.41 | 3.34 | 2.68 | 7.73 |
| Dde_ 0583 | RnfG subunit | 2.83 | 6.35 | 5.19 | 5.43 | 10.07 |
| Dde_ 0584 | RnfE subunit | 1.86 | 4.95 | 3.80 | 3.39 | 7.63 |
| Dde_ 0585 | RnfA subunit | 2.24 | 5.78 | 4.23 | 2.98 | 8.41 |
| Dde_ 0586 | RnfB subunit | 2.18 | 4.39 | 4.16 | 1.48 | 8.09 |
| Dde_ 0587 | RnfF subunit | 2.20 | 4.90 | 5.25 | 1.94 | 10.15 |

Seventeen mutants were obtained that were deficient in syntrophic growth. Of these, two mutants had the transposon insertions within a putative *rnf* operon. This operon is composed of 8 genes located on the genome as shown in Table 3 and genes appear to be co-transcribed. They are separated from the nearest upstream transcribed region by 185 bps and from the nearest downstream transcribed region by 342 bps. The two syntrophy mutants had transposon insertions within *rnf*A and *rnf*D as determined using Arbitrary PCR (*Krumholz et al., 2015*). Both genes encode integral membrane proteins likely located in the cytoplasmic membrane. RnfD has been previously shown to bind a flavin (*Biegel et al., 2011*).

## Transcriptional analysis of the *rnf* operon

The complete transcriptional analysis for strain G20 under lactate-sulfate, $H_2$-sulfate and under syntrophic growth conditions has been previously published (*Krumholz et al., 2015*). Here we present a summary of normalized values for transcription of each subunit within the *rnf* operon (Table 3). Transcription of all genes in the operon including *dchA* are similar under each condition and are affected similarly by growth condition. The most highly expressed genes are the first two in the operon, the gene for the cytochrome c family protein and *rnf*C, both of which have 1.5–2 fold higher levels of expression than genes for the other subunits. Expression was also enhanced approximately 2-fold when cultures were grown on $H_2$, relative to lactate, indicating the importance of this protein for $H_2$ dependent growth. Syntrophic cultures had enhanced expression of all eight genes compared to lactate/sulfate growth with similar levels of expression in the *S. aciditrophicus* co-culture to $H_2$-grown cells and higher levels of expression observed in the *S. wolfei* coculture, indicating the importance of this protein complex under syntrophic conditions. Similar effects on transcription provide evidence that all eight genes are located in one operon.

## Expression analysis of *rnf* operons in mutants

To determine whether the transposon insertion eliminated downstream expression of genes in the *rnf* operon, gap analysis was performed where all intergene regions were amplified except the region between *rnf*B and *rnf*A using cDNA prepared using mRNA.

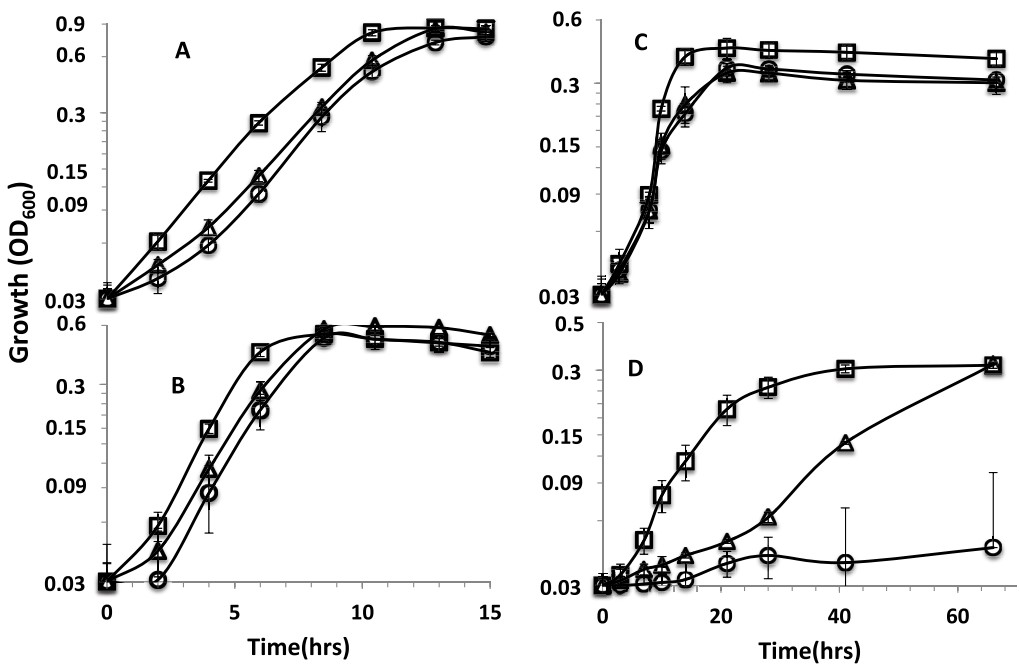

**Figure 1  Growth curves of parent strain and mutants.** Growth curves on (A) lactate-sulfate (B) lactate-sulfite (C) pyruvate-sulfate and (D) formate-sulfate. Curves are for *D. alaskensis* parent strain (□ ) *rnf*A (○) and *rnf*D (△) mutants. Error bars show standard deviation.

**Table 4  Gene expression in *D. alaskensis* *rnf*A and *rnf*D mutants relative to the parental strain using RT-qPCR.** Results were normalized with the 16S rRNA gene.

| Mutant type | Dde_ 587 | Dde_ 589 | Dde_ 590 |
| --- | --- | --- | --- |
| *rnfA* | 0.227±0.255 | 0.608±0.097 | 0.589±0.253 |
| *rnfD* | 1.955±0.624 | 1.155±0.078 | 0.972±0.420 |

All of the intergene regions were amplified for the parent and the two mutants (Fig. S1) confirming that all 8 genes form an operon and indicating that the mutants likely have intact transcripts of the interrupted operon. In this operon, the insertions are likely only affecting the one interrupted gene.

We carried out RT-PCR analysis of the terminal gene in the operon, Dde_ 587 (*rnfF*) as well as the two genes that followed that operon, Dde_ 589 (*uspA*) and Dde_ 590 (*cls*) to be certain that downstream expression was not affected by the insertions. Expression of the terminal gene in the operon was impacted in the mutants with a 4 fold decrease in expression in the *rnfA* mutant and a twofold increase in the *rnfD* mutant (Table 4). The following two genes were minimally impacted indicating that the insertions present in the mutants likely do not affect downstream expression.

### Growth and sulfate reduction by *rnf* mutants

The two mutants exhibited similar growth rates to that of the parent strain when lactate was the electron donor with sulfate (Fig. 1A) or sulfite as electron acceptor (Fig. 1B).

When pyruvate was the donor, growth yields were reduced for the mutants (Fig. 1C). We previously reported that *rnf* mutants grew poorly (*rnf*D) or not at all (*rnf*A) on $H_2$ (*Krumholz et al., 2015*) and a recent study has reported that *rnf* mutants grow poorly or not at all with sulfate as the electron acceptor on malate, fumarate, ethanol, hydrogen, and formate, but growth is not affected in lactate and pyruvate (*Price et al., 2014*). Here, it is shown that the *rnf*D mutant has a long lag phase with formate as the donor, but eventually grew to a similar OD to the parent strain (Fig. 1D). The *rnf*A mutant did not grow on formate or $H_2$ clearly indicating the involvement of Rnf in formate/$H_2$-dependent growth. We also confirmed that the *rnf*A mutant will not grow on ethanol (Fig. S3).

### Sulfide production by washed cells

Washed cells were used to determine if the role for *rnf* was linked to biosynthesis, as mutants were able to grow better on more complex carbon compounds. Incubations of washed cells of the parent strain and the *rnf*A mutant produced 0.57 and 0.17 $\mu$mol sulfide.hr$^{-1}$.mg$^{-1}$ protein, respectively, with lactate as electron donor. Addition of either 5 $\mu$M TCS to the washed cells of the parent strain prevented 95% of the sulfide production and addition of 20 $\mu$M to washed cells inhibited sulfide production by 100%, suggesting TCS directly disrupts the proton gradient needed for respiration. With $H_2$ as the electron donor, parent strain cells produced 0.30 $\mu$mol sulfide.hr$^{-1}$.mg$^{-1}$ protein whereas the *rnf*A mutant did not have any detectable activity. To further test whether the lack of growth on $H_2$ was due to the mutant's inability to biosynthesize carbon intermediates, we attempted to grow the mutant and the parent strain with $H_2$ as the electron donor with 0.1% Casamino acids added to the media. The mutant would still not grow (Fig. S4), providing further evidence that the role for RNF was not directly linked to biosynthesis.

### Growth curves with ionophores

The sodium ion ionophore ETH 2120, tested at 20 $\mu$M, did not have a significant influence on the growth of the parent strain or the *rnf* mutants with lactate as the electron donor whether sulfate or sulfite were present as electron acceptors (Figs. S5 and S6). Interestingly, the protonophore, TCS , at 5 $\mu$M differentially inhibited cultures (Fig. 2). TCS partially inhibited the growth of the parent strain and completely inhibited the growth of *rnf* mutants on lactate-sulfate (Fig. 2A and Fig. S5). Growth on lactate sulfite was then tested and 5 $\mu$M TCS was shown to have a smaller but still significant inhibitory effect on growth in the parent strain (Fig. 2B) and again completely inhibited the *rnf*A mutant (Fig. 2B and Fig. S6). A higher level of TCS was then tested and it was shown that 20 $\mu$M TCS almost completely inhibited the growth of the parent strain on lactate-sulfite (Fig. 2B) and again completely inhibited the mutants (Fig. S7).

### Measurement of $\Delta\Psi$, $\Delta$pH and PMF

To determine whether the Rnf complex is involved in the formation of a proton gradient, the transmembrane potential ($\Delta\Psi$), $\Delta$pH and proton motive force (PMF) were measured in cells of the G20 parent strain and the *rnf*A mutant growing on lactate-sulfate. The $\Delta\Psi$ and $\Delta$pH were both reduced in the mutant (Table 5), which led to a significantly lower PMF in the *rnf*A mutant relative to the G20 parent strain. The lower $\Delta$ pH and PMF in the

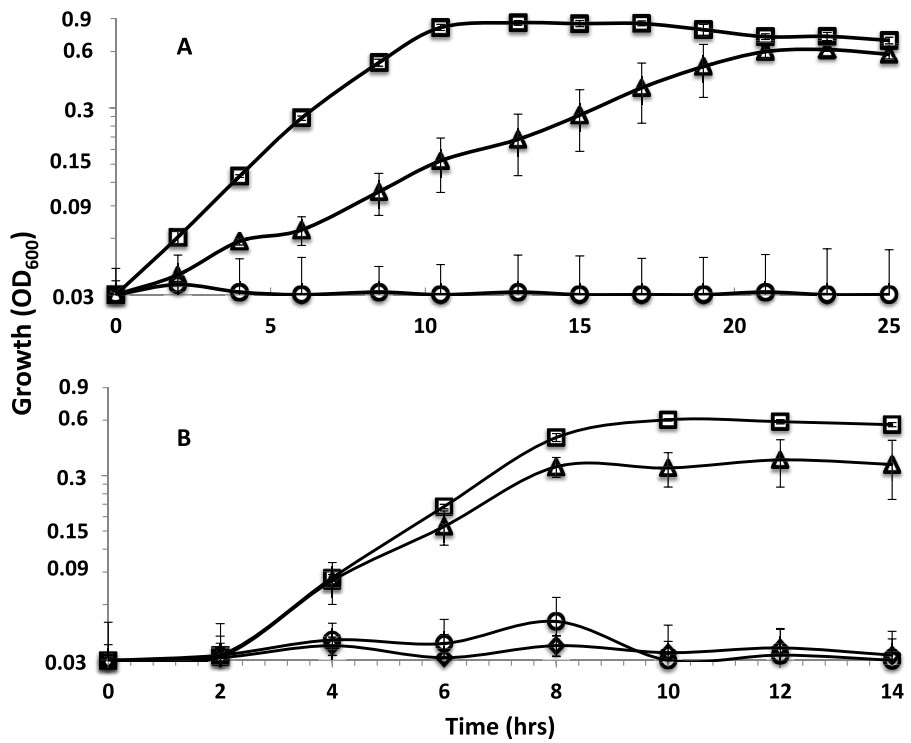

**Figure 2** **Effect of TCS on parent strain and mutants.** Growth curves with (A) lactate-sulfate and (B) lactate sulfite for *D. alaskensis* parent strain no addition (□) and with 5 μM (△) or 20 μM (◇) TCS. Growth curve of the *rnf*A mutant with 5 μM TCS (○) is also shown. Error bars show standard deviation.

**Table 5** **Magnitude of the ΔpH, ΔΨ and total proton motive force of the *D. alaskensis* G20 parent strain and the *rnfA* mutant growing on lactate and sulfate.** Standard deviation in parentheses. Value for the *rnfA* mutant were statistically different from the parent strain at $p < 0.05$.

| Measurement | Parent strain | *rnfA* mutant |
|---|---|---|
| ΔΨ | −158 mV (2.1) | −117 mV (3.2) |
| ΔpH | 0.43 (0.057) | 0.29 (0.015) |
| PMF | −185 mV (3.1) | −135 mV (4.1) |

*rnfA* mutant shows that the Rnf complex clearly had an impact on the energy conservation system by affecting the proton gradient.

## DISCUSSION

The Rnf complex has previously been shown to catalyze the oxidation–reduction reaction between ferredoxin and NAD$^+$ (*Biegel, Schmidt & Müller, 2009*; *Boiangiu et al., 2005*). The redox potential of bacterial ferredoxin can vary, but has been reported within the range of −385 to −460 mV (*Smith & Feinberg, 1990*). That is more negative than that of NAD$^+$/NADH couple (E′ = −280 mV) and therefore the transfer of electrons by Rnf from reduced ferredoxin to NAD$^+$ releases sufficient energy to generate either a proton or a sodium ion potential (*Buckel & Thauer, 2013*). Given that recent studies have

demonstrated both proton (*Tremblay et al., 2013*) and Na$^+$ translocation abilities (*Biegel & Müller, 2010*), it was imperative to determine which ion Rnf translocates in strain G20. Growth experiments showed that lactate-sulfate grown cells were insensitive to the Na$^+$ ionophore, ETH2120, (Figs. S5 and S6) but were highly sensitive to the protonophore, TCS (Fig. 2). Resting cells were also shown to be highly sensitive to TCS suggesting that the proton gradient is needed for sulfate reduction. A similar growth profile was observed in *C. ljungdahlii*, for which this result was interpreted to infer that a proton gradient was needed for growth (*Tremblay et al., 2013*).

When grown on lactate-sulfate or lactate-sulfite, TCS partially inhibited the growth of the parent strain and completely inhibited the growth of *rnfA and rnfD* mutants (Fig. 2). The stronger effect on the mutants suggested that TCS at 5 μM was partially dissolving the proton gradient in the parent strain and this was confirmed when cells were shown to be completely inhibited at 20 μM TCS. We would therefore expect that at 5 μM TCS, growth processes requiring additional proton motive force (or ATP synthesis) would be more highly inhibited than those requiring less ATP. The use of sulfate as an electron acceptor initially requires energy to activate sulfate to adenosine-5'-phosphosulfate (*Gavel et al., 1998*). The requirement for energy to activate sulfate may explain why lactate-sulfate grown cells are more susceptible to the action of TCS than are lactate-sulfite-grown cells as the proton gradient generated during sulfate respiration would be needed to make ATP for sulfate activation. The inability of *rnf* mutants to grow in the presence of 5 uM TCS is consistent with its role in the generation of a proton motive force. In fact, the magnitude of the proton motive force in *rnf* mutants is much less than that in the parent strain of G20 (Table 5).

Ideally, we would have generated complemented *rnf* mutants to prove that the observed insertions were not having polar effects on other genes. We have attempted to clone the *rnfA* and *rnfD* genes into *Escherichia coli* as the first step in complementing the mutants. Unfortunately we have not been successful. Another group has experienced similar problems with *rnf*AB of *Clostridum ljungdahlii* and suggested that *rnf* genes may be toxic to *E. coli* in some cases (*Tremblay et al., 2013*). Gap analysis was used to show that the insertions did not block transcription of downstream genes providing some evidence that polar effects are not important. Also, we carried out RT-PCR analysis of downstream genes. There was a decreased level of expression of the terminal gene in the Rnf operon (*rnfF*) by the *rnfA* mutant, however, there was little effect of the insertions (mutations) on expression of genes downstream of the operon.

Experiments reported here and elsewhere (*Price et al., 2014*) show that the *rnf* mutants are unable to grow on H$_2$, formate and ethanol. These results point to a critical role for Rnf during growth on the above substrates and are consistent with a higher expression level of *rnf* genes when growing with H$_2$ and sulfate relative to lactate and sulfate (Table 3).

A proton gradient is thought to be generated during H$_2$ metabolism in *Desulfovibrio* (*Badziong & Thauer, 1980*) and used for the synthesis of ATP. Membrane vesicle experiments carried out in our lab in an attempt to demonstrate the generation of an ion gradient coupled to the oxidation of reduced ferredoxin and reduction of NAD$^+$ have been unsuccessful. The protein product of the decaheme cytochrome that precedes the

*rnf* operon has been proposed to accept electrons from hydrogenases and shuttle them to Rnf (*Matias et al., 2005*; *Pereira et al., 2011*). However, mutants in this gene had no effect on fitness during growth experiments on ethanol, formate or $H_2$ (*Price et al., 2014*). This suggests that Rnf is not likely receiving electrons directly from $H_2$. It is more likely that *D. alaskensis* relies extensively on ferredoxin oxidation by Rnf to produce a proton gradient during growth on substrates that do not yield net ATP by substrate-level phosphorylation. For those substrates that do yield ATP by substrate level phosphorylation such as malate, fumarate, pyruvate and lactate, a decreased growth rate and or yield was observed in most cases for *rnf* mutants (Fig. 1) (*Price et al., 2014*) suggesting that both Rnf and the F1Fo ATPase are involved in generating a PMF under those conditions.

We are not familiar with any studies describing mechanisms of ferredoxin reduction in *Desulfovibrio*, however, several possible mechanisms have been suggested (*Pereira et al., 2011*; *Price et al., 2014*). During $H_2$ oxidation, these include a possible cytoplasmic electron-bifurcating hydrogenases-linked to a heterodisulfide reductase for which mutants grow poorly on $H_2$ and formate (Hdr/flox-1). For ethanol oxidation, the acetaldehyde:ferredoxin oxidoreductase could be used, and with pyruvate and lactate oxidation, would involve pyruvate:ferredoxin oxidoreductase.

Results from this study are consistent with the fact that *D. alaskensis* Rnf complex functions as a proton rather than a sodium pump and is essential for growth on substrates that do not involve ATP synthesis by substrate-level phosphorylation. Mutation of Rnf limits the development of the PMF and, thus, affects ATP synthesis during growth.

### Funding
This work was supported by a grant from the Physical Biosciences program, DOE Office of Basic Energy Sciences, Chemical Sciences, Geosciences and Biosciences Division. The funders had no role in study design, data collection and analysis, decision to publish, or preparation of the manuscript.

### Grant Disclosures
The following grant information was disclosed by the authors:
Physical Biosciences program.
DOE Office of Basic Energy Sciences, Chemical Sciences, Geosciences and Biosciences Division.

### Competing Interests
The authors declare there are no competing interests.

### Author Contributions

- Luyao Wang and Chuang Li conceived and designed the experiments, performed the experiments, analyzed the data, wrote the paper, prepared figures and/or tables, reviewed drafts of the paper.
- Peter Bradstock conceived and designed the experiments, performed the experiments, analyzed the data, reviewed drafts of the paper.
- Michael J. McInerney conceived and designed the experiments, contributed reagents/materials/analysis tools, wrote the paper, reviewed drafts of the paper.
- Lee R. Krumholz conceived and designed the experiments, performed the experiments, analyzed the data, contributed reagents/materials/analysis tools, wrote the paper, prepared figures and/or tables, reviewed drafts of the paper.

## Data Availability

All of the data has been presented in figures and tables in the manuscript or in the Supplemental Information.

## Supplemental Information

Supplemental information for this article can be found online at http://dx.doi.org/10.7717/peerj.1919#supplemental-information.

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
