# Peer review of "The role of Rnf in ion gradient formation in Desulfovibrio alaskensis"

_PeerJ, doi:10.7717/peerj.1919_

## Round 0.1 · original submission · Major Revisions

The manuscript entitled "Rnf functions in generation of a proton gradient in Desulfovibrio alaskensis" has been reviewed by three independent reviewers. Based on their comments, this manuscript cannot be accepted in its current form due to a number of points raised by the reviewers (considered as major revision). According to these comments, the authors need to provide more experimental data and rewrite certain sections, edit extensively. We encourage you to resubmit this manuscript.

Reviewer 1 ·

Basic reporting

no comments

Experimental design

see below

Validity of the findings

see below

Additional comments

Wang et al have addressed a very important question: is the Rnf complex involved in energy conservation in the sulfate reducer Desulfovibrio alaskensis? To this end, they have generated two rnf mutants, analyzed expression of rnf genes under different growth conditions in wild type and the mutants, analyzed sulfite production in cell suspensions of wt and mutants, explored the effect of ionophores on growth of wt and mutants and established the energetic parameters in wt and mutants.

The paper is generally very well written, but I have problems with the novelty of some findings and with the interpretation of the data.

1. That the rnf mutants do not grow on hydrogen has been reported before by the same group. The new finding is that growth on lactate is not affected by the mutation but that on pyruvate to a small extent.
2. The next paragraph on Sulfide production by washed cells is interesting and novel. I wonder why the ionophore studies have not been done with resting cells. This would be much more convincing that using growing cells. This should be done.
3. To delineate the role of a certain membrane protein or even an ion from inhibitor studies with growing cells is dangerous. For example, inhibition of growth by TCS can be due to eliminating H+-dependent transport systems for, i. e. amino acids. Yes, the authors compared wt and mutants and showed that the mutants are more susceptable but at 20 uM TCS (not a high concentration!) wt and the mutants are inhibited. The only point you can make from that is that the mutants are more susceptible to TCS. One can not state that the Rnf complex is involved in bioenergetics! For that one would need more direct evidence: proton translocation in cell suspensions, respiratory control measurements in cell suspensions and so on. The data one gets out of growing cells are difficult to interprete!
4. The next section Measurement of ΔΨ, ΔpH and PMF is very interesting and this reviewer appreciates the efforts taken by the group to experimentally determine these important parameters. I agree with the interpretations. However, I would like to see how the unspecific binding of TPP was determined. It is not clear to me what is meant by the „nigericin“ experiment, nigericin does not uncouple.

In sum, it is clear that the rnf mutants have a lower pmf and that they are more susceptible at low concentrations to a protonophore. This is, at best, consistent with the hypothesis that the Rnf complex is involved in energy conservation, but no evidence! The paper would be strengthened by adding more data on energetic coupling in cell suspensions.

In the discussion, the authors focus on a role of Rnf in energy conservation when discussing roles of Rnf in metabolism of certain substrates. I don’t find a single reference that describes a role of Rnf as machine for driving uphill transport of electron from NADH to ferredoxin, driven by the electrochemical ion potential. This has been demonstrated experimentally in A. woodii (Hess et al) and shown to be essential for lactate oxidation (Weghoff et al). Is there any chance that Rnf does the same in Desulfovibrio on certain subtrates?

·

Basic reporting

In this study, Wang et al. report that the Rnf complex of the sulfate-reducing bacterium D. alaskensis functions as a proton pump important for growth with several electron donors. The study relies on the characterization of two mutant strains with a transposon insertion in either rnfA or rnfD, two genes probably coding for subunit of the Rnf complex. Overall, the study provides valuable information to the reader about the function of the Rnf complex in energy conservation by SRB. However, there is two major issues and several minor problems with the manuscript that must be addressed before publication.

Major issues:
1-The authors were unable to construct expression plasmids to complement the rnfA and rnfD mutants and demonstrate that the observed phenotypes were really due to the inactivation of those genes. The authors explained why in the discussion and alternatively developed a gap analysis method to demonstrate that intact rnf operon transcripts are still generated by the mutants. However, the authors do not present quantitative comparison between the two mutants and the parent strain and thus do not exclude the possibility that the transposon in rnfA or rnfD could significantly reduce the transcription of downstream genes without completely eliminating it. In my opinion, this experiment needs to be done quantitatively and results must be presented in this study.

2-Numerical values are presented in the main text without standard deviation and number of replicate. Figure 1 and 2 have no number of replicate and standard deviation. This must be addressed. In the current state of the manuscript, it is not clear if most of the results presented by the authors are reproducible or not.

Minor comments:
1-Pg 4 line 67: The hypothesis that the rnf complex of C. ljungdahlii is a proton pump was discussed in details by Kopke et al., 2010 (PNAS). This article should also be discussed here.

2-Pg 4 line 78-79: There is a period missing. More details would be appreciated here. Is the H2 an electron shuttle between Synthrophomonas and Desulfovibrio?

3-Pg 5 line 102: delete one period.

4-Pg 5 line 105: replace “5 uM, 20 uM” by “5 uM or 20 uM”.

5-Pg 5 line 110: replace “rnfA, and rnfB” by “rnfA, and rnfB mutants”

6-Pg 6 line 132: replace “protein” by “protein concentration”

7-Pg 10 line 205-212: See major issue #1

8-Pg 11 line 221: replace “long lag” by “long lag phase”

9-Pg 11 line 224: Where are the results for ethanol growth curves. I would suggest adding it as supplementary information.

10-Pg 11 line 228-230: Standard deviation and number of replicate are missing (see major issue #2)

11-Pg 11 line 235: Where are the results for growth in the presence of casamino acids and H2? I would suggest adding it as supplementary information.

12-Pg 11 line 236: Results for the growth of the rnfD mutant in the presence of the protonophore are not described here or in the figure although there is sentence about that in the abstract (pg 2 line 37-39).

13-Pg 11 line 239: Instead of data not shown, it would be better to add it as a supplementary information.

14-Pg 12 line 246: remove one period.

15-Pg 15 line 315 to 317: A line or two about candidate genes coding for bifurcating hydrogenases or hydrogenases-linked heterodisulfite reductase in D. alaskensis should be included.

16- For Figure 1 and 2 there is not standard deviation and no number of replicate.

17-For table 3 there is no number of replicate.

Experimental design

See basic reporting

Validity of the findings

See basic reporting

Additional comments

See basic reporting

Reviewer 3 ·

Basic reporting

In this study the authors provide results supporting the generation of proton gradient by Rnf membrane protein complex, which had been widely accepted to function as a sodium pump. The article includes sufficient background information to introduce the reader to the relevance of the article.

Experimental design

Following suggestions are recommended for incorporation:
1. Page 5, line 110: ‘rnfA, and rfD using the Qiagen……………..’. Please clarify whether these refer mutants or wild type strains.
2. Page 9, line 193: 'normalized values'. Please clarify which gene has been used for normalization
3. Page 9, line 194: Please clarify the relevance underlying usage of apbE for gene expression comparison purpose.
4. Page 10, line 213 and page 11 line 236: In addition to representing OD as parameter for growth measurement, it is suggested to also include total protein content as a parameter for growth measurement.
5. It is suggested that for all growth experiments standard error bars be included in the graphs.

Validity of the findings

As mentioned previously it is suggested that the growth experiments be in triplicates and results of these should preferably be included as SE bars in graphs.
Rest of the findings are well reported and help in answering the original question asked.

Annotated reviews are not available for download in order to protect the identity of reviewers who chose to remain anonymous.

---

## Round 0.2 · Minor Revisions

Dear Authors,

Your manuscript has been re-reviewed by the reviewers and based on their comments I assign your manuscript as a "minor revision" state. Please note that the first reviewer has raised several critical comments (and also indicated that your manuscript suffers from over statement) that you need to address. These comments are critical as it may require a change in the abstract and title of your manuscript also.

I am hopeful that you will be able to address all these comments accordingly.

Reviewer 1 ·

Basic reporting

see below

Experimental design

see below

Validity of the findings

see below

Additional comments

Wang et al submitted a revised version version of their manuscript. I am afraid to say that my conerns have not been allayed. Still, the only point you can make from the study is that the mutants are more susceptible to TCS. One can, of course not state that the Rnf complex is involved in generation of a proton potential! Yes, the PMF is decreased in the rnf mutants, but only decreased. As correctly stated by the authors, sulfate transport requires energy (as do many other cellular processes required for growth) and therefore, inhibition of growth by a protonophore (or other uncouplers) is expected. As I have pointed out in my previous report, the authors must:
(i) Measure fd oxidation or NAD reduction as a measure for Rnf activity
(ii) next, the effect of the ionophore on any or both of the two redox reactions must be measured and of course, one would expect a stimulation of the reaction. (At the side: the authors see an inhibition of sulfate and sulfide reduction by TCS in washed cells; again this is expected since the electron acceptor has to be transported into the cells and activated. Therefore, this assay can not be used to determine respiratory control! The fact that sulfate reduction (or sulfite reduction ) is inhibited by TCS is NO argument for Rnf being an ion pump!!

A suggestion: If it turns out to be not possible at present time to do the experiments suggested, the statements that Rnf is an ion or even proton pump should be toned down to a level: "that is consistent with...." with Rnf being involved in generation of a H+ gradient". But then the authors should also clearly discuss that there is an additional, apparently major PMF generator.

Specific points:

1. Line 31: incorrect statement, in many bacteria (like E. coli or Rhodobacter) ferredoxin is not involved. Moreover, in these the Rnf complex operates in the reverse direction. Please rephrase.
2. Line 38: not substantiated by the data presented.
3. Line 48-49: No. The authors did not report at all on any biochemical data, it was only speculated that the rnf genes encode a membrane-bound enzyme complex.
4. Line 70: Reference Hess et al, PeerJ
5. Line 74-77: confusing and in conflict with data in Tables. Dos the complex have six or eight subunits?? Please clarify. What is the evidence that Dde_0580 and 0587 are part of the operon or the protein?
6. Line157: exponential growth phase, not logarithmic
7. Line 161-176: Valinomycin only „uncouples“ in the presence of high (200 mM) K+ concentrations. At low concentration, they have the opposite effect. Please state explicitly the K+ concentration in the assay.
8. Line 173-175: Defining the values obtained in the presence of an uncoupler as „unspecific binding“ can of course not be done. Assume, that your uncouplers do not work, than you end up with values that are much too low. The correct way is to use chemcials like butanol or a detergent that are proven to eliminate the potential.
9. Line 251-251: dont see this. The first four data points are actually the same, no difference in growth rate. The mutant seem to go into the stationary phase a bit earlier but whether this is of physiological significance is doubtful.
10. Line 287-294: To include a TCS and ETH experiment here is mandatory!
11. Line 287-294 and discussion: if Rnf activity contributes only to a small extent (only 50 mV!!) to the PMF, where does the rest come from? What is the major PMF generator then?
12. Line 289: incorrect, redox potential of ferredoxin may be much lower than that and actually differs in different organisms.
13. Line 308: A proton gradient can not be needed for respiration. It is generated by respiration or may be nedeed for ATP synthesis.
14. Line 322-325 not substantiated by the data.
15. Line 364-365 not substantiated by the data
16. Line 360-: apparently no proofreading of the references

·

Basic reporting

See general comments for the author.

Experimental design

See general comments for the author.

Validity of the findings

See general comments for the author.

Additional comments

The authors modified and resubmitted their manuscript entitled: “Rnf functions in generation of a proton gradient in Desulfovibrio alaskensis”. In general, the modifications and the supplementary experiments improve the study presented in this manuscript.

I have only several minor comments:

1) Pg 6 line 122: rnfA and rnfD should be italicized.

2) Pg 6 line 124: 4 C should be replaced by 4 °C.

3) Pg 7 line 141: 95 C should be replaced by 95 °C.

4) Pg7 line 142: 60 C should be replaced by 60 °C.

5) Pg 7 line 142-143: (Etarget)ar_target(control-treated)/ (Ereference)Ct_reference (control-treated) should be (Etarget)Ct_target(control-treated)/ (Ereference)Ct_reference (control-treated). For clarity, I would also replace treated by mutant.

6) Pg 11 line 242: For clarity and simplicity, apbE should be replaced everywhere in the main text and tables by rnfF. The homology between both genes is already described in the introduction at pg 4 lines 76-77.

7) Pg 13 line 291: rnfA should be italicized.

8) Pg 14 line 301: The + sign of NAD+ should be superscript.

9) Pg 15 Lines 334-335 in the discussion: This is not exact. Transposon insertion in rnfA has a significant polar effect on the expression of rnfF reducing it by 4 fold. Thus, observations attributed to the inactivation of rnfA could be due to lower expression of rnfF and also of rnfB. This is not a major issue since it does not affect the main conclusion of the authors about the function of the Rnf complex has a primary proton pump in D. alaskensis. Still, I suggest that the authors reformulate this sentence to consider the polar effect of the rnfA mutation on rnfB and rnfF.

10) Pg 16 line 344: The + sign of NAD+ should be superscript.

11) References should be revised: For example, a reference by Tremblay et al., 2012 is included entitled: “A genetic system for Geobacter metallireducens…”. I did not find any relation with the main text.

Reviewer 3 ·

Basic reporting

No comments

Experimental design

• Adequate additional information provided have been provided by the authors in the revised manuscript to address the concerns raised
• Also the authors have provided experimental data to strengthen few aspects of their study.

Validity of the findings

• The authors have substantiated the findings and supported additional information wherever required.

---

## Round 0.3 · accepted · Accept

Dear Authors,

Your revised version is reviewed by two independent reviewers and both have expressed satisfaction on the revised version. Based on their comments, I am happy to inform you that your manuscript has been accepted in PeerJ. Our production team members will contact you in future regarding the publications.

Reviewer 1 ·

Basic reporting

no comment

Experimental design

no comment

Validity of the findings

no comment

Additional comments

The changes have almost adaequately addressed my concerns

·

Basic reporting

See General comments for the author

Experimental design

See General comments for the author

Validity of the findings

See General comments for the author

Additional comments

The manuscript was revised satisfactorily by the authors.
I have no further comments.